# Exact solution of three dimensional schrödinger equation with power function superposition potential

**Meihuan Fu** [ID]\*, **Yongwen Liu, Jianxin Shi** [ID], **Pengbo Qian**

Public Foundational Courses Department, Nanjing Vocational University of Industry Technology, Nanjing, China

\* 1075659384@qq.com

**Data Availability Statement:** All relevant data are within the paper and its Supporting Information files.

**Funding:** The author(s) received no specific funding for this work.

## Abstract

Finding an analytical solution to the Schrödinger equation with power function superposition potential is essential for the development of quantum theory. For example, the harmonic oscillator potential, Coulomb potential, and Klazer potential are all classed as power function superposition potentials. In this study, the general form of the power function superposition potential was used to decompose the second-order radial Schrödinger equation with this potential into the first-order Ricatti equation. Furthermore, two forms of the power function superposition potential are constructed with an exact analytical solution, and the exact bound-state energy level formula is obtained for these two potentials. Finally, the energy levels of some of the diatomic molecules were determined through calculation. And our results are actually consistent with those obtained by other methods.

## Introduction

The potential in the form of $V(r) = a_i r^i (i = 0, \pm1, \pm2, \cdots)$ is known as power function potential. When $i = 0, -1, -2, \cdots$, $V(r)$ is also referred to as inverse power function potential. Playing an important role in quantum mechanics, power function potential is conducive to the study on the structure of some microscopic particles and the interaction between them. For example, harmonic oscillator potential and Coulomb potential are categorized into power function potentials. When more complex microscopic particles are studied, a single power function potential is unable to meet the requirements. However, a better effect can be achieved if the potential function is the superposition of several power functions. The potential in the form of $V(r) = \sum_{i=-n}^{n} a_i r^i$ is called the power function superposition potential, which has attracted attention from many researchers [1–7] looking for the analytical solution of the Schrödinger equation with the power function superposition potential and the energy level of the quantum system. It plays an important role in promoting the development of the quantum theory and its application.

In this study, the general form of the power function superposition potential $V(r)$ is used to decompose the second-order Schrödinger equation with this potential into the first-order Ricatti equation. Then, two forms of power function superposition potential with shape

**Competing interests:** The authors have declared that no competing interests exist.

invariance are constructed, namely, precise analytical solution, and their energy eigenvalues are calculated by using the supersymmetric quantum mechanics (SQM) [8–15].

The remainder of this paper is organized as follows. In the Materials and methods Section, two power function superposition potentials with exact analytical solutions are constructed. The shapes of these two potentials for several different diatomic molecules are presented. The exact bound state energy level formula of these diatomic molecules is obtained. In the Results and discussion Section, the energy eigenvalues of these diatomic molecules are calculated for different radial quantum numbers ($n_r$) and angular momentum quantum numbers ($l$), and compare the results obtained with other methods [16,17]. Finally, a conclusion is drawn in the Conclusion Section.

## Materials and methods

### 1. Construction of two power function superposition potentials by SQM

The radial equation in a central field is expressed as

$$\left[\frac{1}{r}\frac{d^2}{dr^2}r + \frac{2\mu}{\hbar^2}(E - V(r)) - \frac{l(l+1)}{r^2}\right]R_l(r) = 0 \tag{1}$$

where $R_l(r)$ represents the radial wave function, $\mu$ indicates the reduced mass, and angular momentum quantum number $l = 0, 1, 2, \ldots$. Given that $\chi_l(r) = rR_l(r)$, then

$$\frac{d^2\chi_l(r)}{dr^2} + \left[\frac{2\mu}{\hbar^2}(E - V(r)) - \frac{l(l+1)}{r^2}\right]\chi_l(r) = 0 \tag{2}$$

The potential function $V(r)$ is taken as power function superposition, i.e.

$$V(r) = a_{-4}r^{-4} + a_{-3}r^{-3} + a_{-2}r^{-2} + a_{-1}r^{-1} + a_0 + a_1r + a_2r^2 + a_3r^3 + a_4r^4 = \sum_{i=-4}^{4} a_ir^i \tag{3}$$

where the power range of $r$ is just from—4 to 4. When the range is from—6 to 6 or from—8 to 8, the results are consistent.

When the radial quantum number $n_r = 0$, it is assumed that the wave function satisfies the following eigenequation, and the eigenvalue is 0. Therefore,

$$D_-(l)\chi_{0,l}(r) = \left[\frac{d^2}{dr^2} - V_-(r,l)\right]\chi_{0,l}(r) = 0 \tag{4}$$

where

$$V_-(r,l) = \frac{l(l+1)}{r^2} + \frac{2\mu}{\hbar^2}\left[V(r) - E_{0,l}\right] \tag{5}$$

It can be seen that

$$V_-(r,l) = \frac{\chi_{0,l}''(r)}{\chi_{0,l}(r)} \tag{6}$$

Superpotential is defined as

$$W(r,l) = -\frac{\chi_{0,l}'(r)}{\chi_{0,l}(r)} \tag{7}$$

Also, the operator is defined as

$$A_+(l) = \frac{\mathrm{d}}{\mathrm{d}r} + W(r, l) \tag{8}$$

$$A_-(l+1) = \frac{\mathrm{d}}{\mathrm{d}r} - W(r, l) \tag{9}$$

Therefore, the operator $D_-(l)$ can be obtained as

$$D_-(l) = \frac{\mathrm{d}^2}{\mathrm{d}r^2} - V_-(r, l) = A_-(l+1)A_+(l)$$

The Ricatti equation can be obtained as

$$\begin{aligned}
W^2(r, l) - W'(r, l) = V_-(r, l) &= \frac{l(l+1)}{r^2} + \frac{2\mu}{\hbar^2}[V(r) - E_{0,l}] \\
&= \frac{l(l+1)}{r^2} + \frac{2\mu}{\hbar^2}[\sum_{i=-4}^{4} a_i r^i - E_{0,l}]
\end{aligned} \tag{10}$$

When the trial solution $W(r, l) = B_{-2}r^{-2} + B_{-1}r^{-1} + B_0 + B_1 r + B_2 r^2$, it can be obtained that

$$\begin{cases}
B_{-2}^2 = \frac{2\mu}{\hbar^2} a_{-4} \\[2mm]
B_{-2}B_{-1} + B_{-2} = \frac{\mu}{\hbar^2} a_{-3} \\[2mm]
B_{-1} + B_{-1}^2 + 2B_{-2}B_0 = \frac{2\mu}{\hbar^2} a_{-2} + l(l+1) \\[2mm]
B_{-2}B_1 + B_{-1}B_0 = \frac{\mu}{\hbar^2} a_{-1} \\[2mm]
-B_1 + B_0^2 + 2B_{-2}B_2 + 2B_{-1}B_1 = \frac{2\mu}{\hbar^2}(a_0 - E_{0,l}) \\[2mm]
-B_2 + B_{-1}B_2 + B_0 B_1 = \frac{\mu}{\hbar^2} a_1 \\[2mm]
B_1^2 + 2B_0 B_2 = \frac{2\mu}{\hbar^2} a_2 \\[2mm]
B_1 B_2 = \frac{\mu}{\hbar^2} a_3 \\[2mm]
B_2^2 = \frac{2\mu}{\hbar^2} a_4
\end{cases} \tag{11}$$

Since the coefficient $a_i$ ($i = 0, \pm 1, \pm 2, \pm 3, \pm 4$) of the potential function $V(r)$ is independent of the angular momentum quantum number $l$, it is required that

$$
\begin{cases}
B_{-2} = 0 \\
B_{-1} = -\dfrac{1}{2} - \sqrt{\left(l+\dfrac{1}{2}\right)^2 + \dfrac{2\mu}{\hbar^2}a_{-2}} \\
B_0 = \dfrac{\dfrac{\mu}{\hbar^2}a_{-1}}{-\dfrac{1}{2} - \sqrt{\left(l+\dfrac{1}{2}\right)^2 + \dfrac{2\mu}{\hbar^2}a_{-2}}} \\
B_1 = 0 \\
B_2 = 0
\end{cases}
\quad \text{or} \quad
\begin{cases}
B_{-2} = 0 \\
B_{-1} = -\dfrac{1}{2} - \sqrt{\left(l+\dfrac{1}{2}\right)^2 + \dfrac{2\mu}{\hbar^2}a_{-2}} \\
B_0 = 0 \\
B_1 = \sqrt{\dfrac{2\mu}{\hbar^2}a_2} \\
B_2 = 0
\end{cases}
\tag{12}
$$

It can be seen from above that the power function superposition potential $V(r)$ capable to complete the above transformation takes only the following two forms

$$
V(r) = a_{-2}r^{-2} + a_{-1}r^{-1} + a_0
\tag{13}
$$

or

$$
V(r) = a_{-2}r^{-2} + a_0 + a_2 r^2
\tag{14}
$$

There are exact analytical solutions of three-dimensional Schrödinger equation with these two power function superposition potentials. Among them, the first category such as Eq (13) is effectively Coulomb potential plus an inverse quadratic power function, and the other category such as Eq (14) is in essence harmonic oscillator potential plus an inverse quadratic potential term.

When the radial quantum number $n_r = 0$, their energy eigenvalues are expressed respectively as

$$
E_{0,l} = a_0 - \frac{\mu}{2\hbar^2} \frac{a_{-1}^2}{\left(\frac{1}{2} + \sqrt{\left(l+\frac{1}{2}\right)^2 + \frac{2\mu}{\hbar^2}a_{-2}}\right)^2}
\tag{15}
$$

or

$$
E_{0,l} = a_0 + \sqrt{\frac{\hbar^2}{2\mu}a_2} + \sqrt{\frac{2\hbar^2}{\mu}a_2}\left(\frac{1}{2} + \sqrt{\left(l+\frac{1}{2}\right)^2 + \frac{2\mu}{\hbar^2}a_{-2}}\right)
\tag{16}
$$

## 2. The shape of these two power function superposition potentials

The power function superposition potential is a useful model to explore the properties of diatomic molecules. The coefficients of the potentials such as Eqs (13) and (14) can be calculated using the parameters listed in Table 1 for $O_2$, HCl and CO diatomic molecules. The parameters are derived from references [16–20].

where $D_e$ represents the dissociation energy, $r_e$ denotes the equilibrium internuclear separation and $\mu$ refers to the reduced mass.

When $r = r_e$, $V(r) = -D_e$ and $\frac{dV(r)}{dr} = 0$, the coefficients of the potential function $V(r)$ can be obtained through calculation.

If $V(r) = a_{-2}r^{-2} + a_{-1}r^{-1} + a_0$, it can be obtained that $a_{-2} = D_e r_e^2$, $a_{-1} = -2D_e r_e$, $a_0 = 0$.

**Table 1. Reduced masses and spectroscopically determined properties of various diatomic molecules in the ground electronic state.**

| Parameter | $O_2$ | HCl | CO |
|---|---|---|---|
| $D_e$ (in eV) | 5.156658828 | 4.619061175 | 10.84514471 |
| $r_e$ (in nm) | 0.1208 | 0.12746 | 0.11282 |
| $\mu$ (in amu) | 7.997457504 | 0.9801045 | 6.860586 |

**Table 2. The coefficients of the potential $V(r) = a_{-2}r^{-2}+a_{-1}r^{-1}+a_0$.**

| coefficient | $O_2$ | HCl | CO |
|---|---|---|---|
| $a_{-2}$(in eV nm$^2$) | 0.075249266 | 0.075041506 | 0.138040824 |
| $a_{-1}$(in eV nm) | -1.245848773 | -1.177491075 | -2.447098452 |
| $a_0$ | 0 | 0 | 0 |

**Table 3. The coefficients of the potential $V(r) = a_{-2}r^{-2}+a_0+a_2r^2$.**

| coefficient | $O_2$ | HCl | CO |
|---|---|---|---|
| $a_{-2}$(in eV nm$^2$) | 0.075249266 | 0.075041506 | 0.138040824 |
| $a_2$(in eV/ nm$^2$) | 353.3739493 | 284.3190019 | 852.0462326 |
| $a_0$(in eV) | -15.46997648 | -13.85718353 | -32.53543413 |

If $V(r) = a_{-2}r^{-2}+a_0+a_2r^2$, it can be known that $a_{-2} = D_e r_e^2$, $a_2 = \frac{D_e}{r_e^2}$, $a_0 = -3D_e$.

According to the parameters listed in Table 1, the coefficients of these two potentials can be calculated as shown in Tables 2 and 3, respectively, and their $V(r)-r$ curve can also be drawn as shown in Figs 1 and 2, respectively.

For the first potential, as $r$ approaches zero, $V(r)$ becomes infinite because of internuclear repulsion. When $r$ goes to infinity, $V(r)$ is close to zero, i.e., the molecule is decomposed. For the second potential, when $r$ gets close to zero, it is similar to the first potential. When $r$ increases, the harmonic oscillator potential is dominant.

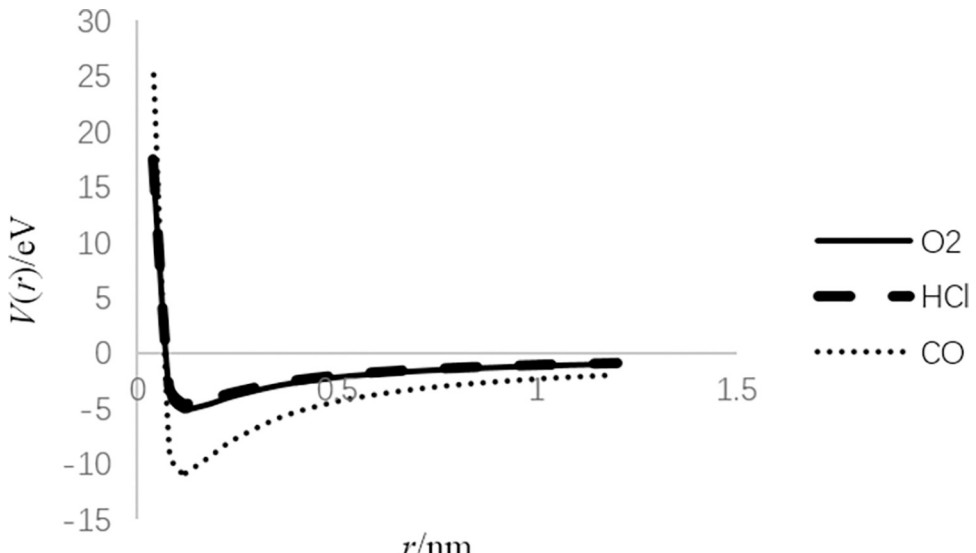

**Fig 1. Shape of $V(r) = a_{-2}r^{-2}+a_{-1}r^{-1}+a_0$ for different diatomic molecules.**

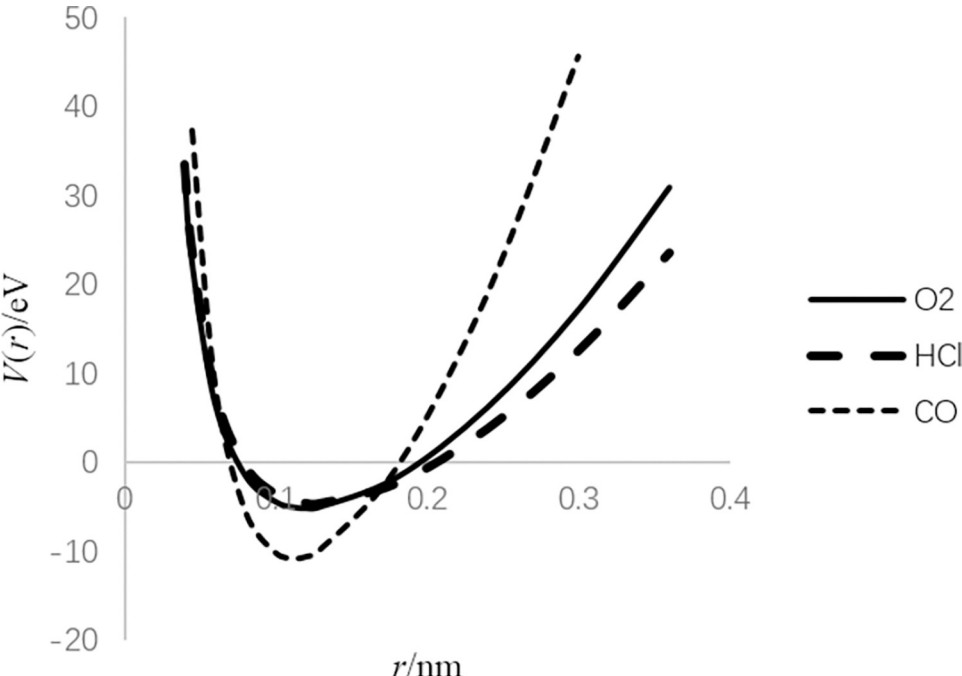

**Fig 2. Shape of** $V(r) = a_{-2}r^{-2} + a_0 + a_2 r^2$ **for different diatomic molecules.**

## 3. Energy levels of the two power function superposition potentials

The energy eigenvalues of the two power function superposition potentials such as Eqs (13) and (14) are calculated according to their shape invariance [21].

**3.1 For the potential** $V(r) = a_{-2}r^{-2} + a_{-1}r^{-1} + a_0$. Let

$$\lambda = \sqrt{\left(l + \frac{1}{2}\right)^2 + \frac{2\mu}{\hbar^2}a_{-2}} - \frac{1}{2} \tag{17}$$

where $\lambda$ refers to the generalized angular momentum quantum number for the potential function $V(r)$. Eqs (8) and (9) can be written as

$$A_+(\lambda) = \frac{d}{dr} - \frac{\lambda + 1}{r} - \frac{\frac{\mu}{\hbar^2}a_{-1}}{\lambda + 1} \tag{18}$$

$$A_-(\lambda + 1) = \frac{d}{dr} + \frac{\lambda + 1}{r} + \frac{\frac{\mu}{\hbar^2}a_{-1}}{\lambda + 1} \tag{19}$$

Since operators $A_-$ and $A_+$ are noncommutative operators, we can make

$$D_+ = A_+(\lambda)A_-(\lambda + 1) = \frac{d^2}{dr^2} - V_+(r, \lambda) \tag{20}$$

It can be obtained that

$$V_+(r, \lambda) = \frac{(\lambda + 1)(\lambda + 2)}{r^2} + \frac{2\frac{\mu}{\hbar^2}a_{-1}}{r} + \frac{\frac{\mu^2}{\hbar^4}a_{-1}^2}{(\lambda + 1)^2} \tag{21}$$

By substituting Eqs (15) and (17) into Eq (5), it can be obtained that

$$V_-(r,\lambda) = \frac{\lambda(\lambda+1)}{r^2} + \frac{\frac{2\mu}{\hbar^2}a_{-1}}{r} + \frac{\frac{\mu^2}{\hbar^4}a_{-1}^2}{(\lambda+1)^2} \tag{22}$$

That is to say,

$$V_+(r,\lambda) = V_-(r,\lambda+1) + \frac{\mu^2}{\hbar^4}a_{-1}^2\left(\frac{1}{(\lambda+1)^2} - \frac{1}{(\lambda+2)^2}\right) \tag{23}$$

If $\alpha_0 = \lambda, \alpha_1 = f(\alpha_0) = f(\lambda) = \lambda+1, R(\alpha_1) = \frac{\mu^2}{\hbar^4}a_{-1}^2\left(\frac{1}{(\lambda+1)^2} - \frac{1}{(\lambda+2)^2}\right)$, then

$$V_+(r,\alpha_0) = V_-(r,\alpha_1) + R(\alpha_1) \tag{24}$$

According to the definition of shape invariance [21], the power function superposition potential in the form of Eq (13) has shape invariance.

If $\alpha_i = f^i(\alpha_0) = \lambda+i$, then $R(\alpha_i) = \frac{\mu^2}{\hbar^4}a_{-1}^2\left(\frac{1}{(\lambda+i)^2} - \frac{1}{(\lambda+i+1)^2}\right)$. According to the definition of shape invariance [21], the energy level of the potential can be expressed as

$$E_{n_r,l} = E_{0,l} + \frac{\hbar^2}{2\mu}\sum_{i=1}^{n_r} R(\alpha_i) = a_0 - \frac{\mu a_{-1}^2}{2\hbar^2(\lambda+n_r+1)^2} \tag{25}$$

where $n_r$ represents radial quantum number.

By substituting Eq (17) into the above formula, the energy level formula of the potential can be obtained as

$$E_{n_r,l} = a_0 - \frac{\mu a_{-1}^2}{2\hbar^2\left(\sqrt{\left(l+\frac{1}{2}\right)^2 + \frac{2\mu}{\hbar^2}a_{-2}} + n_r + \frac{1}{2}\right)^2} \tag{26}$$

### 3.2 For the potential $V(r) = a_{-2}r^{-2}+a_0+a_2r^2$.   Still let

$$\lambda = \sqrt{\left(l+\frac{1}{2}\right)^2 + \frac{2\mu}{\hbar^2}a_{-2}} - \frac{1}{2} \tag{27}$$

where $\lambda$ represents the generalized angular momentum quantum number for the potential function $V(r)$. Eqs (8) and (9) can be written as

$$A_+(\lambda) = \frac{d}{dr} - \frac{\lambda+1}{r} + \sqrt{\frac{2\mu a_2}{\hbar^2}}r \tag{28}$$

$$A_-(\lambda+1) = \frac{d}{dr} + \frac{\lambda+1}{r} - \sqrt{\frac{2\mu a_2}{\hbar^2}}r \tag{29}$$

Considering the noncommutativity of the operators A- and A+, still let

$$D_+ = A_+(\lambda)A_-(\lambda+1) = \frac{d^2}{dr^2} - V_+(r,\lambda) \tag{30}$$

It can be obtained that

$$V_+(r, \lambda) = \frac{(\lambda+1)(\lambda+2)}{r^2} - (2\lambda+1)\sqrt{\frac{2\mu a_2}{\hbar^2}} + \frac{2\mu a_2}{\hbar^2}r^2 \tag{31}$$

By substituting Eqs (16) and (27) into Eq (5), it can be known that

$$V_-(r, \lambda) = \frac{\lambda(\lambda+1)}{r^2} - (2\lambda+3)\sqrt{\frac{2\mu a_2}{\hbar^2}} + \frac{2\mu a_2}{\hbar^2}r^2 \tag{32}$$

That is to say,

$$V_+(r, \lambda) = V_-(r, \lambda+1) + 4\sqrt{\frac{2\mu a_2}{\hbar^2}} \tag{33}$$

Take $\alpha_0 = \lambda, \alpha_1 = f(\alpha_0) = f(\lambda) = \lambda+1, R(\alpha_1) = 4\sqrt{\frac{2\mu a_2}{\hbar^2}}$, then

$$V_+(r, \alpha_0) = V_-(r, \alpha_1) + R(\alpha_1) \tag{34}$$

According to the definition of shape invariance [21], it can be found out that the power function superposition potential in the form of Eq (14) has shape invariance as well.

Let $\alpha_i = f^i(\alpha_0) = \lambda+i$ and $R(\alpha_i) = 4\sqrt{\frac{2\mu a_2}{\hbar^2}}$. According to reference [21], its energy level can be determined through calculation.

$$E_{n_r,l} = E_{0,l} + \frac{\hbar^2}{2\mu}\sum_{i=1}^{n_r} R(\alpha_i) = a_0 + \sqrt{\frac{2\hbar^2 a_2}{\mu}}\left(\lambda + 2n_r + \frac{3}{2}\right) \tag{35}$$

where $n_r$ represents radial quantum number.

Through Eq (27), the energy eigenvalue can be obtained as

$$E_{n_r,l} = a_0 + \sqrt{\frac{2\hbar^2 a_2}{\mu}}\left(\sqrt{\left(l+\frac{1}{2}\right)^2 + \frac{2\mu a_{-2}}{\hbar^2}} + 2n_r + 1\right) \tag{36}$$

## Results and discussion

The energy eigenvalues with Eq (26) can be calculated using the parameters listed in Table 1 for $O_2$, HCl and CO diatomic molecules. The energy eigenvalues are detailed in Table 4 for the different radial quantum number represented by $n_r$ and the angular momentum quantum number denoted as $l$.

Similarly, Table 5 lists the energy eigenvalues obtained by using Eq (36) for $O_2$, HCl and CO diatomic molecules given different radial quantum numbers $n_r$ and angular momentum quantum numbers $l$.

By comparing the data listed in Tables 4 and 6, it can be found out that there are differences in the energy eigenvalues of the same three diatomic molecules as calculated by Eq (26) and Eq (36), and that the difference gradually increases with the rise of quantum numbers, as shown in Table 6. The implications of this are as follows. On the one hand, some potentials may be suitable for study on the characteristics of some microscopic particles, but not for other particles. On the other hand, the effect of harmonic oscillator potential outweighs that of Coulomb potential with the increase of quantum numbers.

Reference [16] used asymptotic iteration method (AIM) to calculate the energy eigenvalues of some diatomic molecules. Reference [17] also calculated the energy eigenvalues of some

**Table 4. Energy eigenvalues (in eV) for the various $n_r$ and $l$ quantum numbers for a few diatomic molecules by using Eq (26), where $\hbar = 1.054571817\times10^{-34}$J·s, $e = 1.602176634\times10^{-19}$C, lamu = $1.66053906660\times10^{-27}$ kg (from SI Brochure 9th edition of the SI Brochure, available on the BIPM web page: www.bipm.org).**

| $n_r$ | $l$ | O₂/eV | HCl/eV | CO/eV |
|---|---|---|---|---|
| 0 | 0 | -5.126358800 | -4.541848670 | -10.79431563 |
| 1 | 0 | -5.066641679 | -4.393729259 | -10.69384082 |
| 1 | 1 | -5.066292858 | -4.391295181 | -10.69337213 |
| 2 | 0 | -5.007961982 | -4.252739178 | -10.59476237 |
| 2 | 1 | -5.007619203 | -4.250421300 | -10.59430017 |
| 2 | 2 | -5.006933786 | -4.245793196 | -10.59337591 |
| 3 | 0 | -4.950295818 | -4.118428126 | -10.49705450 |
| 3 | 1 | -4.949958943 | -4.116219168 | -10.49659869 |
| 3 | 2 | -4.949285330 | -4.111808458 | -10.49568718 |
| 3 | 3 | -4.948275256 | -4.105210348 | -10.49432021 |
| 4 | 0 | -4.893619980 | -3.990380802 | -10.40069206 |
| 4 | 1 | -4.893288873 | -3.988274045 | -10.40024251 |
| 4 | 2 | -4.892626794 | -3.984067324 | -10.39934352 |
| 4 | 3 | -4.891634014 | -3.977774169 | -10.39799534 |
| 4 | 4 | -4.890310938 | -3.969414739 | -10.39619830 |
| 5 | 0 | -4.837911919 | -3.868213688 | -10.30565046 |
| 5 | 1 | -4.837586450 | -3.866202923 | -10.30520706 |
| 5 | 2 | -4.836935644 | -3.862187803 | -10.30432036 |
| 5 | 3 | -4.835959766 | -3.856181097 | -10.30299061 |
| 5 | 4 | -4.834659212 | -3.848201827 | -10.30121815 |
| 5 | 5 | -4.833034511 | -3.838275121 | -10.29900343 |

diatomic molecules using the exact quantization rule method (EQR). For the example of O₂, we compared the energy eigenvalues of its bound states calculated using Eqs (26) and (36) with the energy eigenvalues calculated using other numerical precision methods such as AIM and EQR, as shown in Table 7.

It can be found that the results calculated by Eq (26) are basically the same as those calculated by the AIM and EQR methods. The reason for the slight differences is that some of the parameters we use, such as atomic mass unit (amu), elemental charge ($e$), and reduced Planck constant ($\hbar$), are derived from the latest SI Brochure (9th edition of the SI Brochure, available on the BIPM web page: www.bipm.org), but references [16,17] are not. If the same SI Brochure is used, their calculated results are the same because their potential functions are the same. The significant difference between the results calculated by Eq (36) and other results is due to their different potential functions.

## Conclusion

One of the key tasks of quantum mechanics is to find the exact analytical solution of the Schrödinger equation for any arbitrary $l$ angular momentum quantum number within a given potential. It can be further used to define the observables of the system. In this study, the general form of the power function superposition potential is used to construct two different power function superposition potentials with exact analytical solutions, which is based on the supersymmetric quantum mechanics. The method proposed in this study is a generic one, whose starting point is the general form of power function superposition potential.

Among these two potentials, one such as Eq (13) is actually Coulomb potential plus an inverse quadratic power function, with Kratzer potential falling into this category, while the

**Table 5. Energy eigenvalues (in eV) for the various $n_r$ and $l$ quantum numbers for a few diatomic molecules by using Eq (36), where $\hbar = 1.054571817 \times 10^{-34}$ J·s, $e = 1.602176634 \times 10^{-19}$ C, lamu $= 1.66053906660 \times 10^{-27}$ kg (from SI Brochure 9th edition of the SI Brochure, available on the BIPM web page: www.bipm.org).**

| $n_r$ | $l$ | $O_2$/eV | HCl/eV | CO/eV |
|---|---|---|---|---|
| 0 | 0 | -5.095835179 | -4.463000904 | -10.74318765 |
| 1 | 0 | -4.974277415 | -4.151536686 | -10.53939319 |
| 1 | 1 | -4.973919238 | -4.148911890 | -10.53891450 |
| 2 | 0 | -4.852719651 | -3.840072468 | -10.33559872 |
| 2 | 1 | -4.852361474 | -3.837447672 | -10.33512004 |
| 2 | 2 | -4.851645158 | -3.832200316 | -10.33416269 |
| 3 | 0 | -4.731161887 | -3.528608250 | -10.13180426 |
| 3 | 1 | -4.730803710 | -3.525983454 | -10.13132557 |
| 3 | 2 | -4.730087394 | -3.520736098 | -10.13036823 |
| 3 | 3 | -4.729013012 | -3.512870643 | -10.12893229 |
| 4 | 0 | -4.609604123 | -3.217144032 | -9.928009797 |
| 4 | 1 | -4.609245946 | -3.214519236 | -9.927531108 |
| 4 | 2 | -4.608529630 | -3.209271880 | -9.926573761 |
| 4 | 3 | -4.607455248 | -3.201406425 | -9.925137821 |
| 4 | 4 | -4.606022914 | -3.190929539 | -9.923223382 |
| 5 | 0 | -4.488046359 | -2.905679814 | -9.724215333 |
| 5 | 1 | -4.487688182 | -2.903055018 | -9.723736644 |
| 5 | 2 | -4.486971866 | -2.897807662 | -9.722779297 |
| 5 | 3 | -4.485897484 | -2.889942207 | -9.721343357 |
| 5 | 4 | -4.484465150 | -2.879465321 | -9.719428918 |
| 5 | 5 | -4.482675011 | -2.866385850 | -9.717036107 |

**Table 6. Difference (in eV) of energy eigenvalues calculated by Eq (26) and Eq (36).**

| $n_r$ | $l$ | Difference for $O_2$ | Difference for HCl | Difference for CO |
|---|---|---|---|---|
| 0 | 0 | 0.030523621 | 0.078847766 | 0.05112798 |
| 1 | 0 | 0.092364264 | 0.242192573 | 0.15444763 |
| 1 | 1 | 0.092373620 | 0.242383291 | 0.15445763 |
| 2 | 0 | 0.155242331 | 0.412666710 | 0.25916365 |
| 2 | 1 | 0.155257729 | 0.412973628 | 0.25918013 |
| 2 | 2 | 0.155288628 | 0.413592880 | 0.25921322 |
| 3 | 0 | 0.219133931 | 0.589819876 | 0.36525024 |
| 3 | 1 | 0.219155233 | 0.590235714 | 0.36527312 |
| 3 | 2 | 0.219197936 | 0.591072360 | 0.36531895 |
| 3 | 3 | 0.219262244 | 0.592339705 | 0.36538792 |
| 4 | 0 | 0.284015857 | 0.773236770 | 0.472682263 |
| 4 | 1 | 0.284042927 | 0.773754809 | 0.472711402 |
| 4 | 2 | 0.284097164 | 0.774795444 | 0.472769759 |
| 4 | 3 | 0.284178766 | 0.776367744 | 0.472857519 |
| 4 | 4 | 0.284288024 | 0.778485200 | 0.472974918 |
| 5 | 0 | 0.349865560 | 0.962533874 | 0.581435127 |
| 5 | 1 | 0.349898268 | 0.963147905 | 0.581470416 |
| 5 | 2 | 0.349963778 | 0.964380141 | 0.581541063 |
| 5 | 3 | 0.350062282 | 0.966238890 | 0.581647253 |
| 5 | 4 | 0.350194062 | 0.968736506 | 0.581789232 |
| 5 | 5 | 0.350359500 | 0.971889271 | 0.581967323 |

**Table 7. Comparison of the energy levels (in eV) for the various $n_r$ and $l$ quantum numbers for diatomic molecule $O_2$ calculated using different methods.**

| $n_r$ | $l$ | $O_2$(by Eq 26) | $O_2$(by Eq 36) | $O_2$(by AIM) | $O_2$(by EQR) |
|---|---|---|---|---|---|
| 0 | 0 | -5.126358800 | -5.095835179 | −5.126358625 | −5.126358620071 |
| 1 | 0 | -5.066641679 | -4.974277415 | −5.066641151 | −5.066641146718 |
| 1 | 1 | -5.066292858 | -4.973919238 | −5.066292323 | −5.066292321402 |
| 2 | 0 | -5.007961982 | -4.852719651 | −5.007961116 | −5.007961110233 |
| 2 | 1 | -5.007619203 | -4.852361474 | −5.007618329 | −5.007618327191 |
| 2 | 2 | -5.006933786 | -4.851645158 | −5.006932904 | −5.006932902380 |
| 3 | 0 | -4.950295818 | -4.731161887 | −4.950294624 | −4.950294618656 |
| 3 | 1 | -4.949958943 | -4.730803710 | −4.949957740 | −4.949957739138 |
| 3 | 2 | -4.949285330 | -4.730087394 | −4.949284119 | −4.949284118344 |
| 3 | 3 | -4.948275256 | -4.729013012 | −4.948274034 | −4.948274032620 |
| 4 | 0 | -4.893619980 | -4.609604123 | −4.893618469 | −4.893618463868 |
| 4 | 1 | -4.893288873 | -4.609245946 | −4.893287355 | −4.893287353086 |
| 4 | 2 | -4.892626794 | -4.608529630 | −4.892625268 | −4.892625266816 |
| 4 | 3 | -4.891634014 | -4.607455248 | −4.891632476 | −4.891632475505 |
| 4 | 4 | -4.890310938 | -4.606022914 | −4.890309388 | −4.890309384483 |
| 5 | 0 | -4.837911919 | -4.488046359 | −4.837910103 | −4.837910098245 |
| 5 | 1 | -4.837586450 | -4.487688182 | −4.837584627 | −4.837584625235 |
| 5 | 2 | -4.836935644 | -4.486971866 | −4.836933812 | −4.836933811639 |
| 5 | 3 | -4.835959766 | -4.485897484 | −4.835957923 | −4.835957922172 |
| 5 | 4 | -4.834659212 | -4.484465150 | −4.834657357 | −4.834657353568 |
| 5 | 5 | -4.833034511 | -4.482675011 | −4.833032637 | −4.833032634174 |

other such as Eq (14) is effectively harmonic oscillator potential plus an inverse quadratic potential term. From this, it can be inferred that any potential with an exact analytical solution, plus an inverse quadratic potential term, has an exact analytical solution as well.

Furthermore, the shapes of these two potentials for several different diatomic molecules are presented, and the exact bound state energy eigenvalues of these diatomic molecules are calculated for any $l$ angular momentum quantum number bound by these two exactly solvable potential. The results show that the effect of harmonic oscillator potential is more significant than that of Coulomb potential with the increase of quantum number.

## Supporting information

**S1 Dataset. Minimal data set.**
(DOCX)

## Author Contributions

**Conceptualization:** Meihuan Fu.

**Data curation:** Meihuan Fu, Yongwen Liu.

**Formal analysis:** Meihuan Fu.

**Investigation:** Jianxin Shi.

**Methodology:** Pengbo Qian.

**Writing – original draft:** Meihuan Fu.

**Writing – review & editing:** Meihuan Fu.

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
