## [Decision Letter · Decision Letter 0]

11 Aug 2023

PONE-D-23-15630Exact Solution of Three Dimensional Schrödinger Equation with Power Function Superposition PotentialPLOS ONE

Dear Dr. Fu,

Thank you for submitting your manuscript to PLOS ONE. After careful consideration, we feel that it has merit but does not fully meet PLOS ONE’s publication criteria as it currently stands. Therefore, we invite you to submit a revised version of the manuscript that addresses the points raised during the review process.

We look forward to receiving your revised manuscript.

Kind regards,

Mohammadreza Hadizadeh, Ph.D.

Academic Editor

PLOS ONE

Journal Requirements:

Reviewers' comments:

Reviewer's Responses to Questions

**Comments to the Author**

1. Is the manuscript technically sound, and do the data support the conclusions?

Reviewer #1: Yes

Reviewer #2: Partly

2. Has the statistical analysis been performed appropriately and rigorously? 

Reviewer #1: N/A

Reviewer #2: No

3. Have the authors made all data underlying the findings in their manuscript fully available?

Reviewer #1: Yes

Reviewer #2: No

4. Is the manuscript presented in an intelligible fashion and written in standard English?

Reviewer #1: Yes

Reviewer #2: Yes

5. Review Comments to the Author

Reviewer #1: Is the manuscript technically sound, and do the data support the conclusions? YES. In this work the authors' objective was to develop an analytical solution to the three-dimensional Schrodinger equation. They have successfully derived the necessary equations for two different superposition potentials and solved this to support the conclusions.

Have the authors made all data underlying the findings in their manuscript fully available? YES. The figures are available to download.

Is the manuscript presented in an intelligible fashion and written in standard English? YES. The manuscript is well written, except for the cases where there are capital letters after the equations. These are highlighted in the attachment report.

Reviewer #2: Dear Editor,

In this work, the authors considered a series of the potential but later they have considered only a few terms of the potential and by using of the SUSY formalism they obtained the partner potential and according to the Riccati equation the eigenfunction and eigenvalue of the ground state are obtained and after that the authors applied the results for two cases of the choosing the potential to obtain the energy of the diatomic systems and they reported the results in the two tables and then they compared the results in another tables.There are several serious criticisms that the authors should remove them.

1. The authors never said that the amount of the coefficient constant of the potential is how much and how they obtained these constants.For example the amounts of a_0, a_2 and so on.

2. In table 4 the authors compared the obtained results of two different potentials with each other while we expect the obtained results compared with the experimental data as well as the other methods.

3. There are a lot of typos that should be removed.

4. Introduction should be improved and the related references should be added.

Best Wishes

Hassan

6. PLOS authors have the option to publish the peer review history of their article (what does this mean?). If published, this will include your full peer review and any attached files.

Reviewer #1: **Yes: **Mantile Leslie Lekala

Reviewer #2: No

---

## [Author Response · Author response to Decision Letter 0]

30 Aug 2023

A reply letter to the reviewer

Dear Reviewer,

Thank you very much for your review. Your review comments were very relevant and I was deeply inspired after reading them.

For the following questions, our response is as follows:

1. Is the manuscript technically sound, and do the data support the conclusions?

In our manuscript, the general form of the power function superposition potential is used to construct two different power function superposition potentials with exact analytical solutions, which is based on the supersymmetric quantum mechanics. The exact bound state energy eigenvalues of several different diatomic molecules are calculated for any l angular momentum quantum number bound by these two exactly solvable potential. 

The manuscript must describe a technically sound piece of scientific research with data that supports the conclusion, which is the creed we adhere to when conducting research. 

2. Has the statistical analysis been performed appropriately and rigorously?

Since we construct potential functions with exact analytical solutions and calculate their energy levels, there is no need for statistical analysis. The results provided by the reviewer are correct.

3. Have the authors made all data underlying the findings in their manuscript fully available?

We have provided all the data underlying the findings in our manuscript fully available.

4. Is the manuscript presented in an intelligible fashion and written in standard English?

Our manuscript is presented in an intelligible fashion and written in standard English. Thank you for the reviewer's comments.

5. Review Comments to the Author

We have already introduced in our manuscript how to obtain the number of coefficient constants for the potentials. The original text is as follows:

When... , and , the coefficient of the potential function can be obtained through calculation.

If... , it can be obtained that ... . 

If... , it can be known that ... . 

We did not list the number of coefficient constants for these potentials in our manuscript, and we have added them in the revised manuscript.

Regarding the problem in Table 4, we did compare the results of two different potentials, and we hope to obtain the result that as the quantum number increases, the influence of the harmonic oscillator potential exceeds that of the Coulomb potential. The results obtained were not compared with experimental data or other methods.

In the revised manuscript, we have made modifications and compared our results with other methods to obtain better results.

There are indeed several issues with uppercase letters in our manuscript, and we have made modifications.

We have made improvements to the issues in the introduction and references.

Kind regards,

Meihuan Fu

---

## [Decision Letter · Decision Letter 1]

9 Nov 2023

Exact Solution of Three Dimensional Schrödinger Equation with Power Function Superposition Potential

PONE-D-23-15630R1

Dear Dr. Fu,

We’re pleased to inform you that your manuscript has been judged scientifically suitable for publication and will be formally accepted for publication once it meets all outstanding technical requirements.

Kind regards,

Mohammadreza Hadizadeh, Ph.D.

Academic Editor

PLOS ONE

Additional Editor Comments (optional):

Reviewers' comments:

Reviewer's Responses to Questions

**Comments to the Author**

1. If the authors have adequately addressed your comments raised in a previous round of review and you feel that this manuscript is now acceptable for publication, you may indicate that here to bypass the “Comments to the Author” section, enter your conflict of interest statement in the “Confidential to Editor” section, and submit your "Accept" recommendation.

Reviewer #1: All comments have been addressed

Reviewer #2: All comments have been addressed

2. Is the manuscript technically sound, and do the data support the conclusions?

Reviewer #1: Yes

Reviewer #2: Yes

3. Has the statistical analysis been performed appropriately and rigorously? 

Reviewer #1: (No Response)

Reviewer #2: Yes

4. Have the authors made all data underlying the findings in their manuscript fully available?

Reviewer #1: Yes

Reviewer #2: Yes

5. Is the manuscript presented in an intelligible fashion and written in standard English?

Reviewer #1: Yes

Reviewer #2: Yes

6. Review Comments to the Author

Reviewer #1: The authors have addressed all the recommended changes. The implementation of the changes means the manuscript is now scientifically sound and readable for publication.

Reviewer #2: Dear Editor,

I think the current version is suitable for publication in PLOSE ONE.

Best Wishes.

7. PLOS authors have the option to publish the peer review history of their article (what does this mean?). If published, this will include your full peer review and any attached files.

Reviewer #1: No

Reviewer #2: No

---

## [Editor Report · Acceptance letter]

15 Nov 2023

PONE-D-23-15630R1 

Exact Solution of Three Dimensional Schrödinger Equation with Power Function Superposition Potential 

Dear Dr. Fu:

I'm pleased to inform you that your manuscript has been deemed suitable for publication in PLOS ONE. Congratulations! Your manuscript is now with our production department. 

Kind regards, 

on behalf of

Dr. Mohammadreza Hadizadeh 

Academic Editor

PLOS ONE